# Analytical Determination of Nusselt Numbers for Convective Heat Transfer Coefficients in Channel Macroporous Absorbers

**DOI:** 10.3390/ma17112738

**Published:** 2024-06-04

**Authors:** Andrii Cheilytko, Peter Schwarzbözl, Robin Tim Broeske

**Affiliations:** German Aerospace Center (DLR), Institute of Solar Research, Langenbroich 13, 52428 Julich, Germany

**Keywords:** macroporous media, forced convection, heat transfer coefficient, Nusselt number, modeling of absorbers, high-temperature solar receivers

## Abstract

This article introduces a novel analytical equation for computing the Nusselt number within the macroporous structures of channel absorbers utilized in high-temperature solar receivers. The equation incorporates heat and mass transfer processes occurring within boundary layers as fluid flows through complex-shaped macroporous absorber channels. The importance of accounting for the length of the thermodynamic boundary layer within channel-type macroporous media when calculating heat transfer coefficients using the Nusselt equation is demonstrated. By incorporating proposed indicators of porosity and flow characteristics, this method significantly enhances the accuracy of heat transfer coefficient calculations for such media. Discrepancies observed in existing calculation relationships and experiments are attributed to the omission of certain proposed values in the Nusselt number for macroporous media. To address this, empirical coefficients for the Nusselt number are derived using statistical methods. The resulting semi-empirical equation is applied to macroporous absorbers in solar receivers. The findings enable more accurate predictions of future absorber characteristics, enhancing their efficiency. The derived equation is successfully validated against numerical data across various geometric structures of absorbers in concentrated solar power plants.

## 1. Introduction

Macroporous channel media are used for various types of absorbers in high-temperature solar receivers. The optimization of such structures requires computational models of heat and mass transfer with sufficient accuracy. However, finding an analytical solution of sufficient accuracy for such structures is of scientific and practical interest.

The effective thermal conductivity $\lambda_{ef}$ and effective heat transfer coefficient for the model of heat and mass transfer in a channel-type macroporous medium are convenient values that generalize the complex processes of heat and mass transfer in a macroporous structure. The heat transfer differential equation for the model of channel homogenization in a macroporous medium is
(1)$$\rho_{m}{\left(c_{p}\right)}_{m}\frac{\partial T}{\partial \tau }+\rho_{f}{\left(c_{p}\right)}_{f}\left(\overset{⃑}{u}\nabla T\right)=\nabla \lambda_{ef}\nabla T+q_{source},$$
where ρ—density;

*c_p_*—isobaric heat capacity;m—an index referring to all macroporous material with fluids;f—an index that refers to the fluid phase;*T*—temperature;*τ*—time;$\overset{⃑}{u}\nabla T$—the rate of temperature change in the elementary fluid volume due to convective flows inside the macroporous structure.*q_source_*—thermal energy of heat sources. It may consist of radiation energy, heat of chemical reactions, heat consumption for melting and evaporation, etc.

In recent studies combining continuum and discrete models [1], the following calculation algorithm for conductive-radiative heat transfer in macroporous media is used: First, the effective thermal conductivity and the effective emissivity are approximated. Then, the Stark number is determined as an explicit function of thermal conductivity, porosity, and nominal diameter. The subsequent step involves finding the effective absorption coefficient. Finally, the effective dissipation coefficient is expressed as an explicit function of the emissivity of the solid phase, porosity, and the effective absorption coefficient. The authors recommend this model for small Stark numbers (the error was about 5%). The Rosseland model is suggested only for Stark numbers greater than 1.

For thermal evaluation, the local thermal equilibrium (LTE) method can be used, which assumes a single temperature field for both the liquid phase and the solid structure of the absorber. There are also continuum models based on the more complex Local Thermal Non-Equilibrium (LTNE) approach. LTNE modeling assumes two temperature fields for the liquid phase and the solid absorber with separate differential equations. The two phases are connected through convective heat transfer.

Quasihomogeneous approximations are one of the most common models for describing transport processes in macroporous media. In this method, the real macroporous medium is replaced by a continuous medium with effective parameters, and the equations valid in a single pore (capillary) are used as macroscopic equations. The heat transfer is modeled using the transient energy equation.
(2)$$\rho_{s}{\left(c_{p}\right)}_{s}\frac{\partial T_{s}}{\partial \tau }=\frac{\partial }{\partial x}\left(\lambda_{ef}\frac{\partial T}{\partial x}\right)-\alpha A_{v}\left(T_{s}-T_{f}\right)+q_{source}$$
where α—convective heat transfer coefficient W/(m^2^∙K).

*s*—an index that refers to the solid phase;*f*—an index that refers to the fluid phase;*A_v_*—the surface area per volume of the absorber structure.

For analytical models of heat and mass transfer, the convective heat transfer coefficient is usually calculated using the Nusselt number.

## 2. Methods of Contemporary Models of Convective Heat Transfer Coefficient in Macroporous Media

According to the LTNE approach, the energy equation for the macroporous material can be written as follows:(3)$$\rho_{s}{\left(c_{p}\right)}_{s}\frac{\partial T_{s}}{\partial \tau }=\frac{\partial }{\partial x}\left(\lambda_{s}\frac{\partial T}{\partial x}\right)-\alpha A_{v}\left(T_{s}-T_{f}\right)+q_{source}$$

The energy equation for the fluid phase is defined as
(4)$$\rho_{f}{\left(c_{p}\right)}_{f}\left(\frac{\partial T_{f}}{\partial \tau }+uT_{f}\right)=\frac{\partial }{\partial x}\left(\lambda_{f}\frac{\partial T}{\partial x}\right)+\alpha A_{v}\left(T_{s}-T_{f}\right)+q_{source}$$

To determine a calculation method for the convective heat transfer coefficient for macroporous channel-type media, stationary fluid flow through a single channel of an arbitrary shape is considered. Equations (3) and (4) can be rewritten as follows for a stationary process:(5)$$\frac{\partial }{\partial x}\left(\lambda_{s\_ef}\frac{\partial T}{\partial x}\right)-\alpha A_{v}\left(T_{s}-T_{f}\right)+q_{source}=0$$
(6)$$\frac{\partial }{\partial x}\left(\lambda_{f\_ef}\frac{\partial T}{\partial x}\right)+\alpha A_{v}\left(T_{s}-T_{f}\right)+q_{source}=0$$
where $\lambda_{s\_ef}$—is the effective thermal conductivity coefficient of the solid phase;

$\lambda_{f\_ef}$—is the effective thermal conductivity coefficient of the fluid phase.

For further consideration of the convective heat transfer coefficient, it is necessary to ensure that in the considered structures, the Knudsen number is much less than unity since it determines the conditions of heat and mass transfer in macroporous material. To meet this condition, the minimum macropore size was considered to be 0.1 mm.

In a previous publication, the authors proposed the following formulas for calculating the effective coefficient of thermal conductivity [2]:(7)$$\lambda_{s\_ef}=\frac{\frac{\lambda_{f}\lambda_{s}}{\lambda_{s}\emptyset +\lambda_{f}\left(1-\emptyset \right)}\left[\frac{\lambda_{s}}{\lambda_{f}}\emptyset +\left(1-\emptyset \right)+\Psi \right]\left[\emptyset +\frac{\lambda_{f}}{\lambda_{s}}\left(1-\emptyset \right)+\Psi \right]}{\frac{\lambda_{s}}{\lambda_{f}}\emptyset +2\Psi +\frac{\lambda_{f}}{\lambda_{s}}\left(1-\emptyset \right)}$$
where for the fluid phase $\Psi_{f}$ is defined as follows:(8)$$\Psi_{f}={\left(k_{y}\frac{1-\frac{\lambda_{f}}{\lambda_{s}}+\sqrt{{\left(1-\frac{\lambda_{f}}{\lambda_{s}}\right)}^{2}+4}}{2}\right)}^{1-\emptyset }\times {\left(k_{y}\frac{1-\frac{\lambda_{s}}{\lambda_{f}}+\sqrt{{\left(1-\frac{\lambda_{s}}{\lambda_{f}}\right)}^{2}+4}}{2}\right)}^{\emptyset }$$
and for the solid phase as follows:(9)$$\Psi_{s}=\frac{1}{k_{z}}\left(\emptyset -1\right)\frac{\left(\lambda_{s}-\lambda_{f}\right)}{{\lambda_{s}\lambda }_{f}}\left[\lambda_{f}\left(\emptyset -1\right)-\lambda_{s}(\emptyset )\right]$$
where ∅ is porosity;

k_x_ and ky is the total radius vector showing the displacement of the channels relative to each other in the x and y planes, respectively (for inline array, it is equal to 1).

In this case, the coefficient $k_{z}$ is considered the displacement of elementary channels in the plane z. The *z*-axis is the axis perpendicular to the channel area under consideration. For repeating unit cells, $k_{z}=1$. For honeycomb structures, $k_{z}=1/sin\gamma$, γ is the offset angle of the next layer of channels.

The convective heat transfer coefficient in macroporous media is analytically found from the Nusselt number. In general form, the equation of heat transfer coefficient for the absorber channel can be written as follows:(10)$$Nu=c{Re}^{n}{{Pr}_{f}}^{\frac{1}{3}}{\left(\frac{{Pr}_{f}}{{Pr}_{s}}\right)}^{0.25},$$
where c,n—are coefficients that consider the location of pores. In [3], it is proposed to consider c and n as a Darcy function.

*Pr*—Prandtl number
(11)$$Pr=\frac{c_{p}\mu }{\lambda }$$
where μ—dynamic viscosity, Pa⸱s

*c_p_*—specific heat, J/(kg·K)

The average method for calculating the heat transfer coefficient given in [4] is
(12)$$\alpha_{v}=l_{m}{\left(\frac{\dot{m}}{d_{h}}\right)}^{l_{n}}$$
where $l_{m}$ and $l_{n}$ are empirical constants depending on the Reynolds number;

*d_h_*—hydraulic diameter;$\dot{m}$—mass flow density.

Here is an analysis of the types of Nusselt number equations and coefficients c and n proposed in the literature. In [5], five different authors are analyzed who propose different heat transfer coefficient correlations for dispersed bulk materials. Bulk materials are considered a macroporous medium with channels formed by voids. The author concludes that for values of $Re\in \left(20;200\right)$




(13.1)
Nu=0.106Re



(13.2)$$\alpha =0.106\frac{\lambda_{f}}{\mu }\dot{G},$$where $\dot{G}$—mass flow of fluid per unit area.

For Re > 200:(14.1)$$Nu=0.61{Re}^{0.67},$$



(14.2)
$$\alpha =0.61\frac{\lambda_{f}}{\mu^{0.67}d^{0.8}}{\dot{G}}^{0.67}$$



In article [6], the heat transfer coefficient of gas passing through macroporous ceramics is calculated from the Nusselt number.
(15)$$\alpha_{v}=\frac{Nu\cdot \lambda_{f}}{\langle d_{pore}\rangle },$$

In article [7], the following models are offered:(16.1)$$\alpha_{v}=a_{sf}\frac{Nu\cdot \lambda_{f}}{\langle d_{pore}\rangle },$$



(16.2)
$$Nu=2.0696\emptyset^{0.38}{Re}^{0.438}$$



(16.3)$$a_{sf}=\frac{10.33H\sqrt{1-\emptyset }-5.8H^{1.6}\left(1-\emptyset \right)}{d}$$where $a_{sf}$—specific surface area;

H = 1 is defined as the Heywood circularity factor

The author in [8] uses the following model for macroporous solar receivers of open type:(17.1)$$\alpha_{v}=\frac{Nu\cdot \lambda_{f}}{{\langle d_{pore}\rangle }^{2}}$$



(17.2)
$$Nu=\left(32.504\emptyset^{0.38}-109.94\emptyset^{1.38}+166.65\emptyset^{2.38}-86.98\emptyset^{3.38}\right){Re}^{0.438}$$



The author in [9] proposes an equation for porous media in solar thermal molten salt energy storage systems.
(18)$$Nu=6\left(1-\emptyset \right)\left(2+1.1{Re}^{0.6}{Pr}^{\frac{1}{3}}\right)$$

One of the seven models considered by the authors in [10] has the most adequate physical explanation and passes the test for limit transitions, namely model [11].
(19)$$Nu=0.34\emptyset^{-2}{Re}^{0.61}{Pr}^{\frac{1}{3}}$$

Using different models of volumetric convective heat transfer coefficients in [10] leads to a difference in efficiency of 27.7%, indicating a high sensitivity of the thermal efficiency of the absorber of the considered models to convective heat transfer coefficients. Based on a comparative analysis of data from different authors, the following formula is recommended for porous absorbers in high-temperature solar receivers [12]:(20)$$Nu=\left(0.0426+1.236\frac{d_{m}}{L}\right){Re}_{d_{m}}$$

A formal pore diameter based on the manufacturer’s specification d = 1/PPC, a ‘passage’ diameter that includes the porosity $d_{m}=\sqrt{4\emptyset \pi }/PPC$, and an effective strut diameter we can find as follows:(21)$$d_{s}=d\cdot \frac{2\left(0.5+cos\left(\frac{1}{3}\arccos\left(2\emptyset -1\right)+\frac{4}{3}\pi \right)\right)}{\sqrt{\pi }}$$

It is worth noting that none of the proposed models considers the Mikheev correction [13] for the variability of the properties of capillary fluids ${\left(\frac{{Pr}_{f}}{{Pr}_{f\_w}}\right)}^{0.25}$. In this correction, the thermophysical properties of the fluid are calculated for the fluid temperature and wall temperature. This correction establishes the relationship between the heat transfer coefficient of the fluid and the wall temperature. Suppose that the average wall temperature can be calculated using the following formula:(22)$$T_{f\_w}=T_{s\_min}\frac{x}{l}+\frac{{T_{s\_max}+T}_{f\_in}}{2}\left(1-\frac{x}{l}\right)$$
where $T_{s\_max}$ and $T_{s\_min}\;$are the maximum temperature and minimum temperature of the solid phase in the absorber, which lies in the area at a distance from the front of the surface of maximum radiation absorption, K;

$\frac{x}{l}\;$is the relative depth of the zone of maximum absorption of solar radiation in the absorber.

Among the amendments, it should be noted that for microporous structures, the authors [14] recommend calculating the actual velocity in microchannels $w_{0}\;$ as the velocity is determined through the mass flow rate, considering the superficial velocity as follows:(23.1)$$w_{s}=\frac{\dot{m}}{\rho_{fw}A}$$
and reference speed for calculations as follows:(23.2)$$w_{0}=\frac{d^{2}∆p}{L\mu }$$where

d—hydraulic diameter, m;L—channel length, m;*ρ_fw_*—fluid density by a wall temperature, kg/m^3^*σ_μ_*—experimental value of the viscous slip coefficient = 1.1

The use of another approach, through the effective coefficient of thermal conductivity and determination of heat flow on computer models of solar receivers, can provide more accurate results, but only for a specific porous structure of the modeled object. In his dissertation, Grobbel J. [15] calculates the convective heat transfer coefficient between dispersed spherical particles as a sequence of two thermal resistances (method from [16]): contact resistance $\frac{1}{\alpha_{WS}}\;$and penetration resistance $\frac{1}{\alpha_{bed}}$, as follows: (24.1)$$\frac{1}{\alpha }=\frac{1}{\alpha_{WS}}+\frac{1}{\alpha_{bed}},$$



(24.2)
$$\alpha_{bed}=\frac{1}{\sqrt{\pi }}\frac{\sqrt{\rho c_{p}\lambda_{bed}}}{\sqrt{t}}$$



(24.3)$$\alpha_{WS}=A_{cont}\frac{4\lambda_{f}}{d_{p}}\left[\left(1+\frac{2\left(l+\delta \right)}{d_{p}}\right)Ln\left(1+\frac{d_{p}}{2\left(l+\delta \right)}\right)-1\right]$$where

*A_cont_*—contact area,*d_p_*—particle diameter,*l*—the modified mean free path and *δ*—the surface roughness of the particles.

When using the channel structure, the equation used for small-diameter pipes was also considered Equations (25.1) and (25.2). The macroporous medium is represented by staggered or corridor pipes. For them, Nu is determined from known equations, and the length of hydrodynamic and thermal stabilization in the channel is determined by
(25.1)$$l_{g}=k_{g}\cdot Re\cdot d$$

(25.2)$$l_{T}=k_{T}\cdot Re\cdot d\cdot Pr$$
where $k_{T}\;and$ $k_{g}$ are empirical coefficients equal to 0.07 and 0.065 for round channels.

Replacing the macroporous medium with a channel medium can be geometrically acceptable for many open-type macroporous media configurations. This method can be applied analogously to the passage of air through pipes. In the article [17], the similarity of the obtained experimental data for the Nusselt number in metal foams for the case of laminar flow inside the pipe, which gives the Nusselt numbers, is noted as Nu ≈ 3.7 (for the defining pore diameter of 0.5 mm, the value Re ≈ 50).

For air flow on top of a bundle of pipes at Re < 1000, take
(26)$$\bar{Nu}=0.49{Re}^{0.5}$$

According to the hydrodynamic theory of heat transfer for the developed turbulent continuous motion, there is a connection between heat transfer and resistance, which can be expressed as follows [13]
(27.1)$$Nu=\frac{\xi }{8}Pe\cdot E,$$

(27.2)$$\alpha =\frac{\xi }{8}c_{p}\rho w\cdot E$$where ξ—aerodynamic resistance

*Pe*—Peclet number
(27.3)$$Pe=\frac{\langle d_{pore}\rangle u}{D}$$

*E*—coefficient



(27.4)
$$E=\frac{1}{1+12\sqrt{\frac{\xi }{8}}\left({Pr}^{2/3}-1\right)}.$$



Having a set of calculation equations for the Nusselt number and several different methods for determining the convective heat transfer coefficient in a macroporous medium (numerical calculation method [18,19], analytical calculation method based on the Nusselt and Reynolds number, Nusselt and Peclet number, analytical method of heat transfer calculation as thermal resistance [15], analytical method of Fourier solution or homotropy analysis method [20], etc.), it can be concluded that an integrated approach is required to determine the convective heat transfer coefficient of macroporous media. At the same time, it is not possible to compare the methods and their contribution to the model of heat transfer in a macroporous medium of channel type for different absorber structures with different coolants in order to determine the best one for all open-type solar absorbers. This is primarily due to the lack of clear mathematical numbers for describing the macroporous medium of channel type. Numerical calculation methods are set only by our idea of heat transfer in a macroporous medium, considering only the general, averaged distribution of thermal energy over the material without considering the physical essence of the heat transfer itself and neglecting microprocesses. However, at the same time, they strictly describe a given geometric characteristic of the macroporous medium of channel type and allow for the determination of a result based on the current level of knowledge about heat transfer processes. Therefore, it is necessary to find a new computational Nusselt equation that will include indicators of heat and mass transfer in the porous medium. To research these indicators, the existing dependencies were analyzed.

To construct a graphical dependence of the Nusselt number on Re for various equations and experimental data (Figure 1), it was assumed that the characteristics of a macroporous medium are constant. The Reynolds number was calculated for the temperature at which most types of absorbers are currently tested, according to [16] ${\bar{T}}_{Pr\_f}=584$ K; Pr = 0.7023; $\lambda_{f}=0.045$ W/(m∙K); and the desired temperature level, ${\bar{T}}_{Pr\_f(des)}=670.33$ K; Pr(des) = 0.708; $\lambda_{f(des)}=0.05$ W/(m∙K). The fluid or void cross-section Af defines the porosity; therefore, it can be directly recalculated from the porosity and the unit cell cross-section: $A_{f}=A\cdot \emptyset$. The solid cross-section is then simply given as $A_{s}=A-A_{f}=A\left(1-\emptyset \right)$.

It was considered that the coolant velocity is 1.5 m/s, the equivalent channel diameter is 2 mm, the porosity of the absorber is 90%, and the value of Re = 92.5 was obtained. Therefore, in Figure 1, the range Re ∈ [5;150] is chosen. The data obtained using a numerical model for high-temperature solar receivers [21] (based on the numerical LTNE method) are listed as the “numerical DLR method”. “Numerical DLR method front” is a Nusselt calculation for the flow perpendicular to the porous medium (used for the surface of a porous absorber). The numerical results are compared against experimental data taken from [14,22,23,24].

**Figure 1 materials-17-02738-f001:**
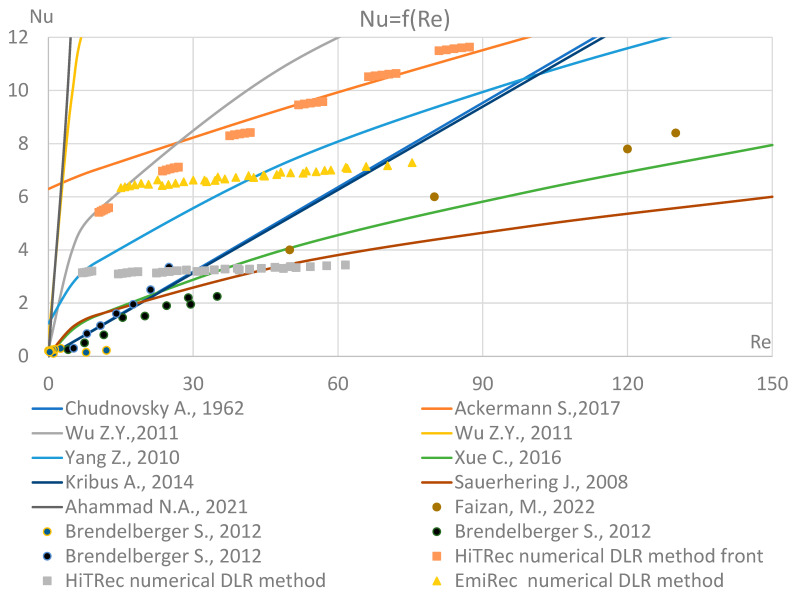
Comparative analysis of the existing dependences of the Nusselt number on Reynolds for macroporous media Re ∈ [5;150]. References to the literature: Chudnovsky A., 1962 [5], Ackermann S., 2017 [6], Wu Z.Y., 2011 [7], Wu Z.Y., 2011 [8], Yang Z., 2010 [9], Xue C., 2016 [11], Kribus A., 2014 [12], Sauerhering J, 2008 [17], Ahammad N.A., 2021 [18], Faizan, M., 2022 [24], Brendelberger S., 2012 [22].

From Figure 1, it can be concluded that for macroporous absorbers of open type, it is most appropriate to use the equations given by [5,11,12,17]. For microporosity with Re < 3, the equation [11,17] can be used. However, it should be noted that the experimental data on various macroporous structures of open-type absorbers are not enough to give priority to a specific equation.

According to [25], the following equation holds as an approximate solution for a hydrodynamically developed laminar flow in a pipe:(28)$${Nu}_{0}=\frac{3.657}{\tanh\left(2.264X^{\frac{1}{3}}+1.7X^{\frac{2}{3}}\right)}+\frac{0.0499}{X}\tanh\left(X\right)$$
(29)$$X=\frac{L}{d_{e}RePr}$$

There is a proposal to find Nu in [25] as follows:(30)$$Nu=\frac{{Nu}_{0}}{\tanh\left(2.43{Pr}^{\frac{1}{6}}X^{\frac{1}{6}}\right)}.$$

However, in our case, this value did not coincide with the results obtained.

Figure 1 shows the uncertainty of experimental data and existing dependencies. Moreover, by analyzing the dependency of the Nusselt number on the Reynolds number, it can be concluded that as the geometric scale of the porosity decreases, the effect of the Reynolds number becomes smaller, i.e., the exponential coefficient of the Reynolds number is close to one (cf., e.g., Equation (10)).

An attempt is made to calculate the Nusselt numbers in the channel of a porous absorber by the classical method of determining the coefficients of dependence of the Nusselt number on the Reynolds number using data from the numerical method [21]. The channel of the porous absorber StepRec2 is shown in Figure 2. This type of absorber was chosen as an example for two reasons: according to the authors, it is the most promising absorber for modern solar towers, and it has a rather complex shape that can be divided into simple sections. Other types of absorbers were also considered by the authors and are briefly described in the following sections:

To show the impossibility of defining the coefficients c and n as constants (Equation (10)), Figure 3 shows the variation in the Nusselt number from Reynolds for the channel front of the StepRec2 absorber (absorber with complex shape channel, Figure 2) calculated using the DLR numerical method.

According to the classical method of experimental determination of the constants included in Equation (10), the dependence of Nu on Re is determined at a constant fluid Prandtl number. It has been previously proven that the correction of capillary fluid ${\left(\frac{{Pr}_{f}}{{Pr}_{s}}\right)}^{0.25}$ has a positive effect on the accuracy of THE power law dependence of the Nusselt number on the Reynolds numbers for the absorber channel. Therefore, Figure 2 shows an approximation of the Nusselt–Reynolds dependency according to classical theory to determine the constants of the equation.

In Figure 3, the Nusselt coefficients can be grouped by absorber section. For a conventional infinite channel, this plot is a continuous step dependence with a constant power number from 0.25 to 0.75 (Equation (10)). In Figure 3, however, a discrete function is observed, which can be divided into separate functions by channel section. In addition, the function for the first segment is also discrete. The obvious difference between the sections is the different hydraulic diameters. However, even if the Nusselt coefficients are calculated separately for each section according to the classical theory, the deviation of the obtained Nusselt numbers from the average one is too large for the first section. Therefore, an innovative approach is needed to construct a single equation that describes all sections of a given absorber simultaneously. Also, sections 4 and 5 of StepRec2 are identical in terms of effective properties such as porosity, specific surface area, etc. They have the same square channel geometry but are offset by half the cell width. The data obtained show that as the velocity increases, the convective heat transfer decreases in short sections. This can be explained by the formation of a hydrodynamic and thermal boundary layer as the fluid flows around the front surface of the macroporous channel. Therefore, it is concluded that it is necessary to recalculate the obtained data considering the thickness of the hydraulic boundary layer.

It is obvious that porosity is not a unimodal criterion for choosing one or another type of dependence for the calculation of thermal processes of convective heat transfer in a macroporous medium. Therefore, in addition to porosity, it is necessary to introduce additional indicators of the porous structure into the calculation equations.

## 3. Finding of Nusselt Number for Convective Heat Transfer Coefficient in Macroporous Absorbers of the High-Temperature Solar Receiver

The function for the Nusselt number in the general case of the macroporous absorber of the air receiver, which most fully describes the ratio of convective to conductive heat transfer in the absorber of the solar receiver, will be
(31.1)$$Nu=c{Re}^{n}{Pr}^{m}{\left(\frac{{Pr}_{f}}{{Pr}_{s}}\right)}^{0.25},$$

Suggest that $c=f\left(\frac{\emptyset }{1-\emptyset }\right)$ considered as an analog of a homogeneous void fraction.
(31.2)$$n=f\left(l_{t},L,d_{e}\right)$$where

*L*—length of the channel;*l_t_*—length of the thermal boundary layer (Figure 3);*d_e_*—hydraulic diameter.

By determining the significant coefficients, it was found that considering the ratio $n=f(\frac{d_{e}}{L})$ increases the accuracy of the approximation of the Nusselt number from the Reynolds numbers for different channel geometries. Also, it was proposed to calculate the constant c as a function of the ratio of the hydraulic diameter to the length of the hydrodynamic layer and the ratio of the section length to the length of the thermodynamic layer.
(31.3)$$C=f\left(\frac{de}{lt},\frac{T_{in}}{T_{w}},\frac{1-\emptyset }{\emptyset }\right)$$

This proposal is an innovative solution and significantly increases the accuracy of the calculation of the heat transfer coefficient in channel macroporous structures. So, the following equation is proposed for macroporous channel structures to calculate the Nusselt number:(32)$$Nu=c\cdot K^{stability}\cdot Exp\left(-0.5\frac{T_{in}}{T_{w}}\right)\cdot \frac{d_{e}}{l_{t}}{\cdot Re}^{n}{{\cdot Pr}_{f}}^{0.33}\cdot {\left(\frac{{Pr}_{f}}{{Pr}_{s}}\right)}^{0.25},\;c=c_{1}\cdot exp\left(c_{2}\cdot tan\left(\frac{1-\emptyset }{\emptyset }\right)\right),\;K^{stability}=\left\{\begin{array}{c} 1\;if\;l_{t}^{previous}<L^{previous}\\ 1-0.1\frac{l_{t}^{previous}-L^{previous}}{L},\\ 0.9\;if\;l_{t}^{previous}>\left(L^{previous}+L\right) \end{array}\right.\;n=0.015Ln\left(\frac{d_{e}}{L}\right)+c_{3}{\left(\frac{c}{\emptyset }\xi_{Shape}\right)}^{c_{4}},\;\xi_{Shape}\approx \frac{d_{e}}{d_{e}^{previous}}+1+\frac{d_{e}}{d_{e}^{next}}.\;l_{T}=k_{T}\cdot Re\cdot d\cdot Pr,\;k_{T}=0.07$$
where $\xi_{Shape}$—coefficient changes the shape of the channel;

for the first section, $\xi_{Shape}=\emptyset +1+\frac{d_{e}}{d_{e}^{next}}$;for the last section, $\xi_{Shape}=\frac{d_{e}}{d_{e}^{previous}}+1-\emptyset$;

$c_{1}=0.1,c_{2}=0.525,$ $c_{3}=1.1,and$ $c_{4}=-0.09$ are structural constants. These coefficients are the same for the entire geometry of the complex channel shape (for all sections of the macroporous channel).

The $c_{i}$ coefficients were selected based on the results of approximation and averaging between absorber sections. The values of the averaged coefficients c and n and the method of their calculation are shown in Table 1. The statistic deviations of the invented formula are shown in Table 2.

The approximation graphs are shown below (Figure 4):

The obtained coefficients are $c_{1}=0.1;c_{2}=0.525;$ $c_{3}=1.1;$ $c_{4}=-0.09$. The statistical errors for the sections of the StepRec2 model for different section lengths are shown in the table. As can be seen, the accuracy of the universal calculation for the StepRec2 model was 18.3%.

Proposed porosity indicators $\frac{de}{lt},\frac{1-\emptyset }{\emptyset },\frac{de}{L}$ are such that they will significantly improve the accuracy of the calculation of the Nusselt number and make it possible to predict its change in macroporous channel structures.

For complex geometries of macroporous channel structures, having only four total coefficients instead of two local ones for each section, as suggested by classical theory (10 coefficients for the absorber used as an example), simplifies the engineering task of finding an equation to solve the Nu number. As will be shown below, the proposed coefficients are also close to universal and can be used with some accuracy for all macroporous channel structures.

## 4. Result and Discussion

The proposed formula was validated using a numerical method on the DLR computer model. The computational model was built in the OpenFOAM Version 6 open-source software environment and included 4164 Nusselt number calculations for various configurations of the StepRec2 absorber, each with different input parameters [21]. Below are validation graphs (Figure 5, Figure 6, Figure 7, Figure 8 and Figure 9) comparing calculations using the proposed formula with those using the standard engineering method outlined in HeatAtlas [16]. The graphs depict results for five different section lengths of the StepRec2 model.

In all cases, the calculation based on the proposed formula provides greater accuracy than the standard method. Furthermore, it demonstrates the potential to reduce the maximum deviation to 10.9% by recalculating coefficients for each channel section based on the hydraulic diameter.

For the first section of the model with manually adjusted coefficients from StepRec2, the maximum deviation is 5.5%. This accuracy is considered sufficient to prevent overfitting of the coefficients.

For the second section of the model with manually adjusted coefficients of StepRec2, the maximum deviation is 7.7%. This accuracy is considered sufficient compared to the model with universal coefficients.

For the third section of the model with manually adjusted coefficients from StepRec2, the maximum deviation is 17%. The values of the coefficients could not be chosen for greater accuracy.

For the fourth section of the model with manually adjusted coefficients of StepRec2, the maximum deviation is 14.5%. The values of the coefficients could not be chosen for higher accuracy.

For the fifth section of the model with manually adjusted coefficients of StepRec2, the maximal deviation is 11.2%. However, the values of these coefficients could not be further optimized for greater accuracy.

The increased accuracy of the proposed formula, as demonstrated in previous graphs, does not alter the functional dependence of the Nusselt number. Therefore, it is proposed to use the newly derived formula with constant coefficients for engineering calculations. This will facilitate the construction of an optimization model for heat and mass transfer in structured channel macroporous media, enabling the identification of a preliminary range of geometric characteristics that maximize thermal efficiency.

Further refinement of the optimal geometric characteristics of the channel-type macroporous medium for more precise calculations will be achieved through the use of numerical models. This approach is expected to significantly reduce both the time and computational resources required to identify the optimal macroporous structure of the absorber.

An additional potential application of the developed formula is in adjusting coefficients to enhance the calculation accuracy for new types of absorbers. The future objective will be to minimize the number of numerical models required for different geometries while maximizing the accuracy of the developed formula.

To provide insight into the variety of porous structures discussed in this article, Figure 10 illustrates the geometric structure of porous absorbers utilized in a solar tower.

The proposed method has also been tested for different types of macroporous absorber structures (existing solar absorbers of optimal configuration) for different input mass fluxes and different temperatures. The results are presented in Figure 11, Figure 12, Figure 13 and Figure 14. The proposed equation with the same coefficients was used for validation with different channel shapes and diameters. The validation results are presented below. The model of Hoffschmidt B. for calculating Nusselt numbers [26] was also used for the validation of HitRec and the first section of EmiRec.

In the figure, the deviation in the first section is from −6.6% to 14.3%. The deviation in the second section is from 2% to 33.7%.

The presented validation for completely different geometries and parameters of fluid confirms the possibility of using the proposed formula for high-temperature solar receivers and shows the importance of using the proposed porous structure indicators for macroporous channel structures.

Understanding the type of flow in macroporous channels is important for using standard similarity models and for understanding the heat and mass transfer processes in the interchannel macroporous medium. Most authors consider the motion in the macroporous medium of the channel type of the absorber of solar tower stations laminar due to low flow velocities [27,28,29]. Indeed, based on the analysis (Figure 3: Comparative analysis of the existing dependences of the Nusselt number on Reynolds for macroporous media), the fluid Reynolds values are too small to consider the flow as turbulent, but the study shows [30] vortices and micro convective reverse currents appear at the non-linear structure of the channel. Therefore, it is proposed to improve the formula for calculating the heat transfer coefficient according to the Nusselt number by introducing an additional variable, the dislocation vector k_y_, which describes the unevenness of the structure [2]. This factor is entered as a correction to the equivalent diameter and takes the value of one for equal channels as follows:(33)$$\alpha_{v}=\frac{Nu\cdot \lambda_{f}}{\langle \bar{d_{e}}\rangle k_{y}k_{z}}.$$

With an analytical solution to the energy transfer equation, it is possible to create an optimization model for maximizing the thermal efficiency of a macroporous absorber used in open-type receivers for solar tower power plants. The obtained equation also shows the possibility of improving the existing calculation dependences of the Nusselt number for macroporous channel-type media by including the stabilization coefficient $K^{stability}$, $Exp\left(-0.5\frac{T_{in}}{T_{w}}\right)\;and\;\frac{d_{e}}{l_{t}}$.

To determine the influence of input factors on heat and mass transfer in porous media, we will conduct the following analysis:(34.1)$$\mathrm{For}\;Nu=c\cdot K^{stability}\cdot Exp\left(-0.5\frac{T_{in}}{T_{w}}\right)\cdot \frac{d_{e}}{l_{t}}{\cdot Re}^{n}{{\cdot Pr}_{f}}^{0.33}\cdot {\left(\frac{{Pr}_{f}}{{Pr}_{s}}\right)}^{0.25}\to max$$



(34.2)
$$\mathrm{have}\;c=0.1\cdot \exp(0.525\cdot \tan(\frac{1-\emptyset }{\emptyset }))\to \max$$



From this relationship, it can be imagined that there is a decrease in the porosity of the medium while maintaining all other geometric parameters. However, the function of the coefficient c is a family of descending curves, and therefore, the porosity is uncertain, and its optimal value depends on other factors.

Consider the factor of flow stabilization
(34.3)$$K^{stability}\to max\to 1\;so\;l_{t}<L$$



(34.4)
therefore, L≥0.07·Re·d·Pr



Consider the influence of other factors included in the coefficient n
(34.5)$$n=0.015Ln\left(\frac{d_{e}}{L}\right)+1.1{\left(\frac{c}{\emptyset }\xi_{Shape}\right)}^{-0.09}\to max$$



(34.6)
$$0.015\mathrm{Ln}(\frac{de}{L})\to \max;$$





(34.7)
therefore, de→max, L→min. L = 0.07·Re·dPr.





(34.8)
$$1.1{\left(\frac{c}{\emptyset }\xi_{Shape}\right)}^{-0.09}\to max$$





(34.9)
$$\xi_{Shape}\approx \frac{d_{e}}{d_{e}^{previous}}+1+\frac{d_{e}}{d_{e}^{next}}\to min$$



for the first section, $\xi_{Shape}=\emptyset +1+\frac{d_{e}}{d_{e}^{next}}\to min$; The porosity of the first section should be maximal with $d_{e}\ll d_{e}^{next}$for the last section, $\xi_{Shape}=\frac{d_{e}}{d_{e}^{previous}}+1-\emptyset \to min$. The porosity of the last section should be minimal with $d_{e}\ll d_{e}^{previous}$

## 5. Conclusions

The proposed porosity indicators $\frac{de}{lt},\frac{1-\emptyset }{\emptyset },\frac{de}{L}$ offer significant potential for enhancing the accuracy of Nusselt number calculations and enabling the prediction of convective heat transfer coefficients in macroporous channel structures. Moreover, the article introduces a novel analytical method for computing heat transfer coefficients using the Nusselt number. This method stands out for its incorporation of additional macroporous medium indicators, such as the stabilization coefficient $K^{stability}$ and the pore shape $\xi_{Shape}$. Additionally, it includes calculations for the lengths of the hydrodynamic and thermodynamic layers within macroporous media with channel-type pores.

The universal formula for calculating the Nu number in the absorber channel is as follows:(35)$$Nu=c\cdot K^{stability}\cdot Exp\left(-0.5\frac{T_{in}}{T_{w}}\right)\cdot \frac{d_{e}}{l_{t}}{\cdot Re}^{n}{{\cdot Pr}_{f}}^{0.33}\cdot {\left(\frac{{Pr}_{f}}{{Pr}_{s}}\right)}^{0.25},\;c=c_{1}\cdot exp\left(c_{2}\cdot tan\left(\frac{1-\emptyset }{\emptyset }\right)\right),\;K^{stability}=\left\{\begin{array}{c} 1\;if\;l_{t}^{previous}<L^{previous}\\ 1-0.1\frac{l_{t}^{previous}-L^{previous}}{L},\\ 0.9\;if\;l_{t}^{previous}>L^{previous} \end{array},\;n=0.015Ln\left(\frac{d_{e}}{L}\right)+c_{3}{\left(\frac{c}{\emptyset }\xi_{Shape}\right)}^{c_{4}},\;\right.\xi_{Shape}\approx \frac{d_{e}}{d_{e}^{previous}}+1+\frac{d_{e}}{d_{e}^{next}}.\;l_{T}=k_{T}\cdot Re\cdot d\cdot Pr,\;k_{T}=0.07$$

In this study, constants for the proposed dependencies for modern types of solar receiver absorbers were determined. The constants of the proposed Nusselt equation were found to be$\;c_{1}=0.1;c_{2}=0.525;$ $c_{3}=1.1;$ $c_{4}=-0.09$.

It was determined that stabilization of the flow in the channel has a positive effect on heat transfer, and for this purpose, the desired length of each section of the channel (without changing the shape) should be greater than 0.07∙Re∙d∙Pr. However, as the channel size increases, the degree of the Reynolds criterion decreases slightly. Therefore, the rational choice of channel length is 0.07∙Re∙d∙Pr.

Increasing the hydraulic diameter of the channel with the same overall porosity of the medium increases the heat transfer of the fluid.

Verification of the results on the DLR numerical model demonstrated satisfactory reliability and a significant improvement in accuracy compared to modern analytical models.

The root mean square error for the determined geometrical and physical parameter variations were as follows: for the StepRec2 type absorber (20,820 variations), it was 0.296627; for the StepRec type absorber (35 physical parameter variations), it was 0.3597; and for the HiTRec type absorber (30 physical parameter variations), it was 0.4184.

For complex geometries of macroporous duct structures, having only four coefficients instead of the traditional ten simplifies the engineering task of solving the Nusselt number equation. This reduction in coefficients streamlines the process without sacrificing accuracy.

The potential applications of the proposed dependence are extensive. It can be used for mathematical optimization of heat and mass transfer processes in macroporous open channel media, allowing the analytical determination of preliminary design parameters without relying solely on numerical methods. Furthermore, it can improve numerical methods for optimizing the structure of macroporous open channel media and reduce iterations in machine learning algorithms.

Overall, the results of this study provide a promising way to improve the efficiency and accuracy of thermal analysis in solar receiver absorbers, paving the way for advances in solar energy technology.

## Figures and Tables

**Figure 2 materials-17-02738-f002:**
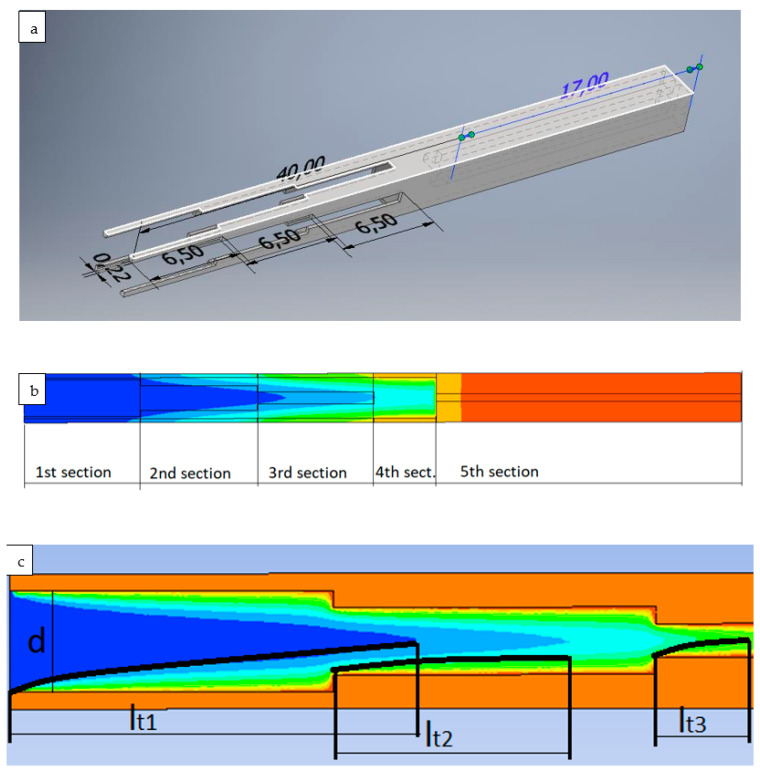
Flow enthalpy distribution over the absorber of StepRec2 with an explanation of thermal boundary layer lengths: (**a**)—a general view of one absorber channel; (**b**)—a 2D side view of the channel divided into its five geometric sections, where the color indicates the fluid temperature (blue is the lowest fluid temperature, red is the highest); (**c**)—a 2D side view of the first two channel sections, where the color indicates the fluid enthalpy in the channel (blue is the lowest value, red is the highest) and the black lines indicate the thermal boundary layer lengths.

**Figure 3 materials-17-02738-f003:**
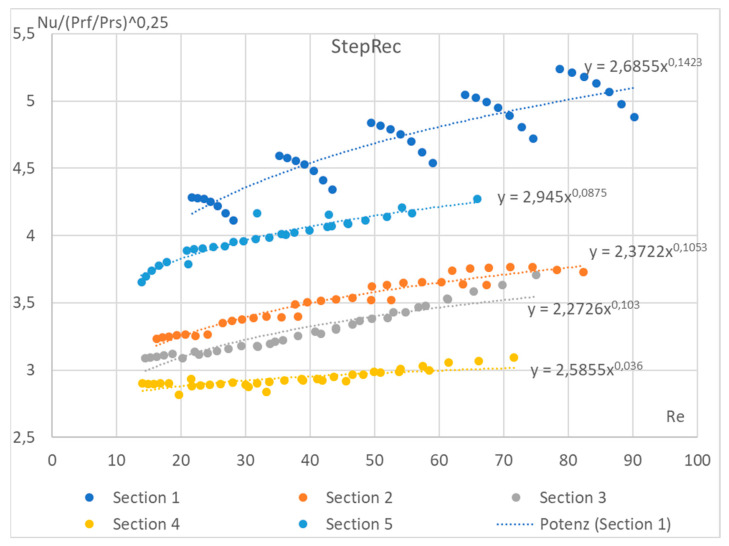
Dependence of the Nusselt number on Reynolds for a complex macroporous channel structure (StepRec2 absorber) based on numerical studies.

**Figure 4 materials-17-02738-f004:**
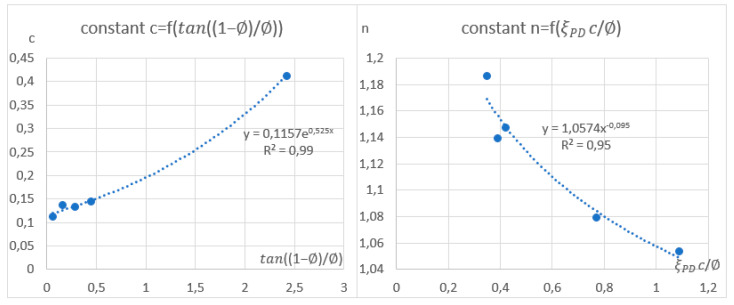
Approximation of the averaged constants of the criterion equation Nu by sections.

**Figure 5 materials-17-02738-f005:**
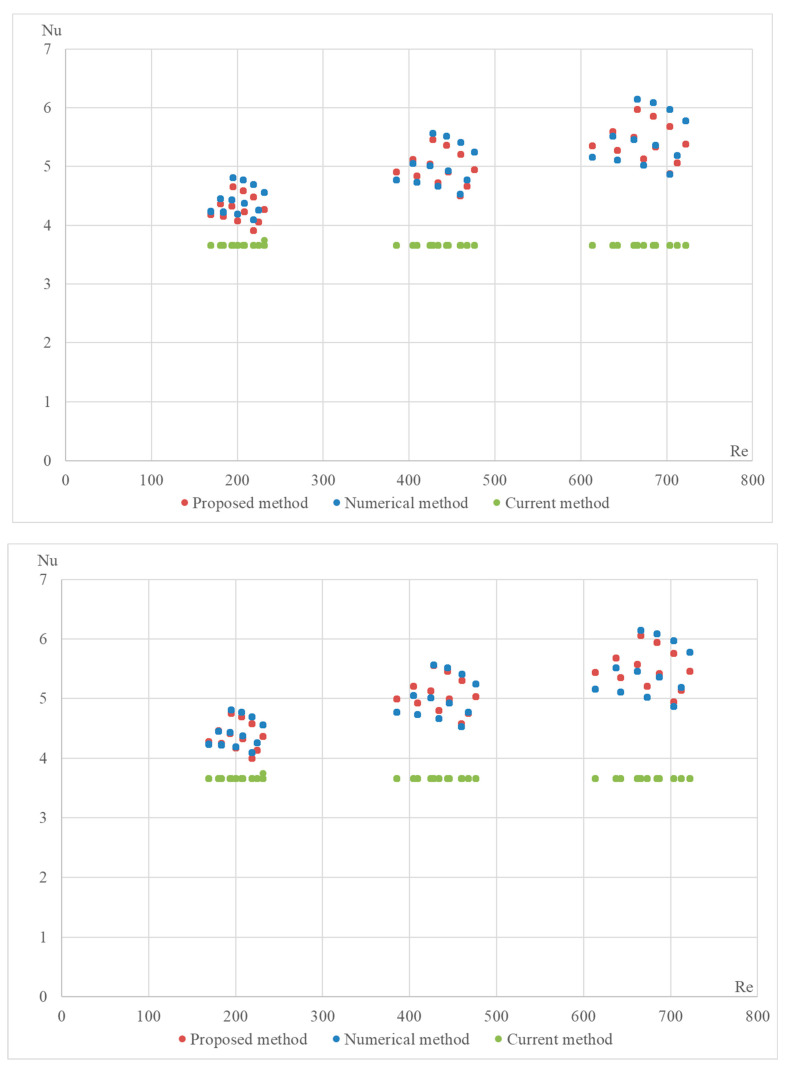
Validation of the Nu number of the dependence on Re for different variations in the model in the first section of StepRec2: model with general coefficients (**top**) and model with manually adjusted coefficients $c_{1}=0.105;c_{2}=0.525;$ $c_{3}=1.1;$ $c_{4}=-0.09$ (**bottom**).

**Figure 6 materials-17-02738-f006:**
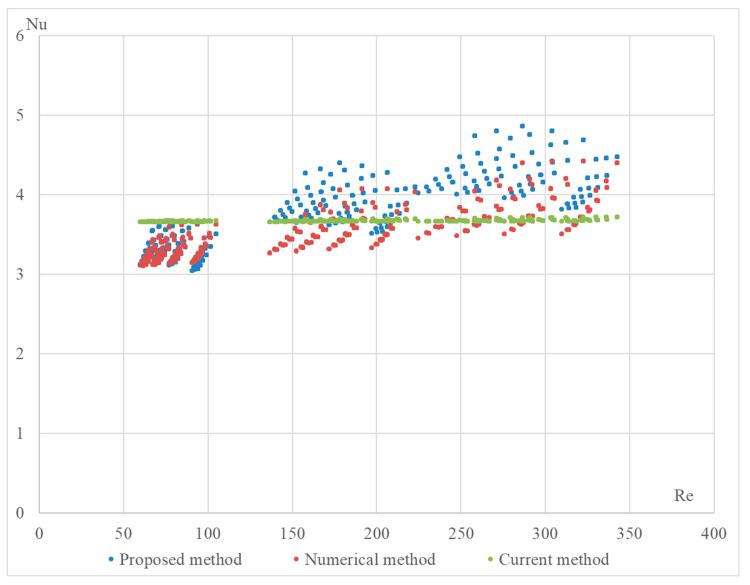
Validation of the Nu number of the dependence on Re for different variations in the model in the second section of StepRec2: model with general coefficients (**top**) and model with manually adjusted coefficients $c_{1}=0.1193;c_{2}=0.525;$ $c_{3}=1.15;$ $c_{4}=0$ (**bottom**).

**Figure 7 materials-17-02738-f007:**
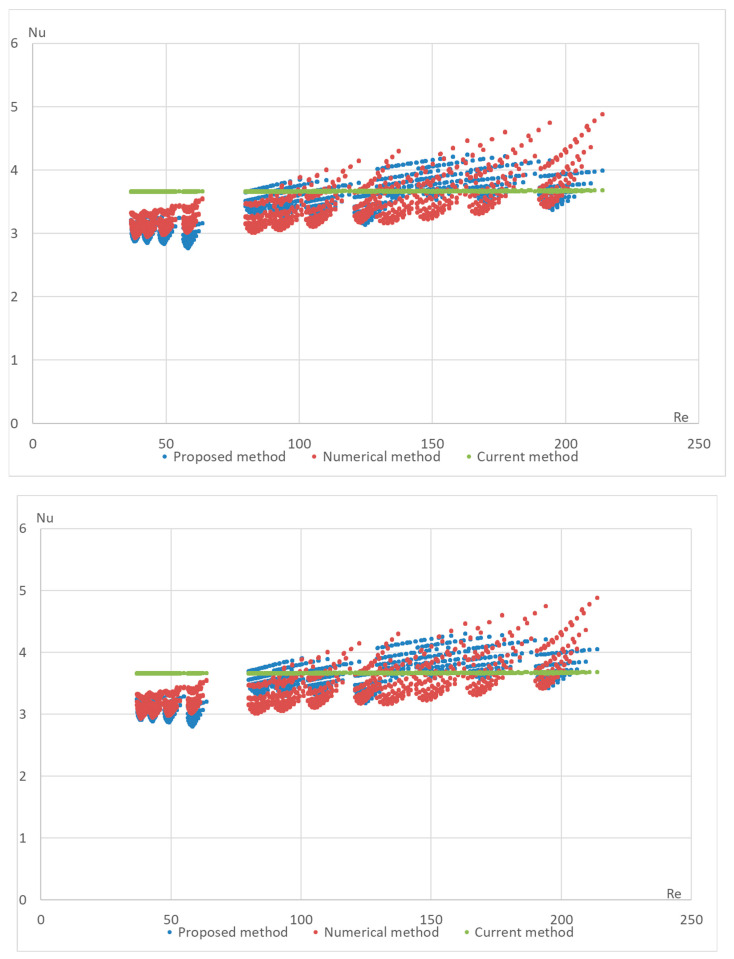
Validation of the Nu number of the dependence on Re for different variations in the model in the third section of StepRec2: model with general coefficients (**top**) and model with manually adjusted coefficients $c_{1}=0.1;c_{2}=0.525;$ $c_{3}=1.08;$ $c_{4}=-0.114$ (**bottom**).

**Figure 8 materials-17-02738-f008:**
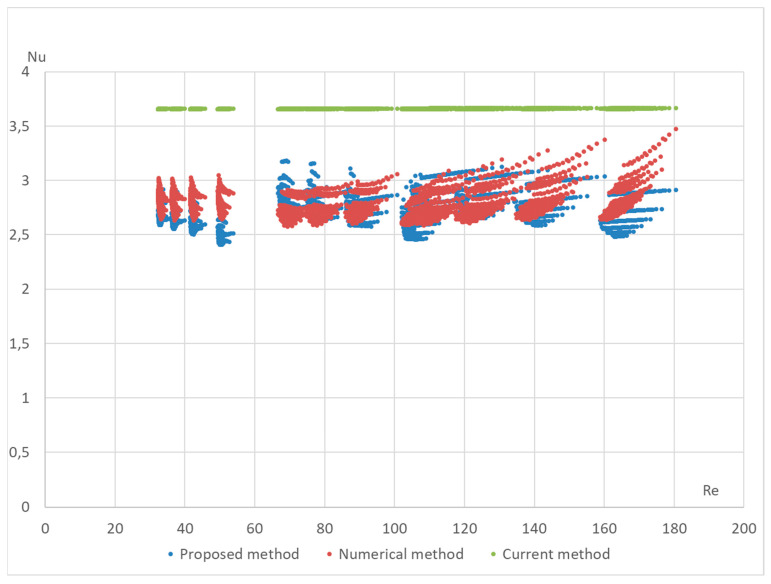
Validation of the Nu number of the dependence on Re for different variations in the model in the fourth section of StepRec2: model with general coefficients (**top**) and model with manually adjusted coefficients $c_{1}=0.104;c_{2}=0.525;$ $c_{3}=1.1;$ $c_{4}=-0.091$ (**bottom**).

**Figure 9 materials-17-02738-f009:**
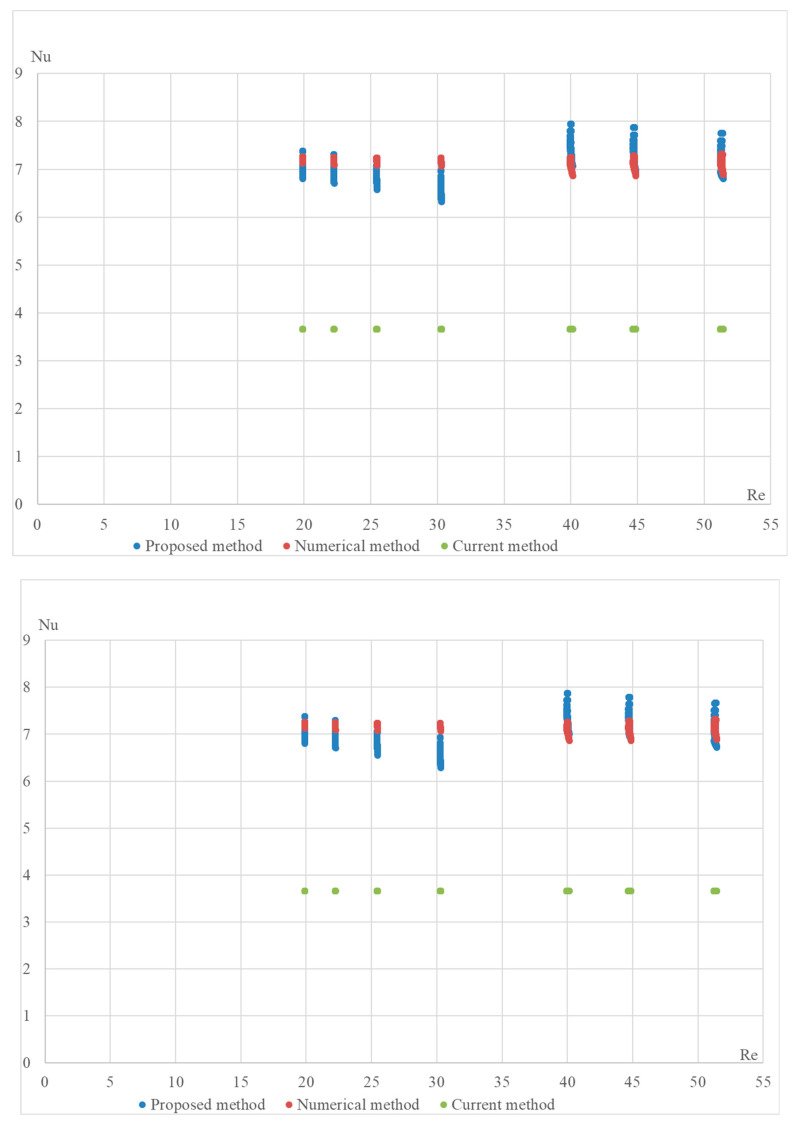
Validation of the Nu number of the dependence on Re for different variations in the model in the fifth section of StepRec2: model with general coefficients (**top**) and model with manually adjusted coefficients $c_{1}=0.0104;c_{2}=0.525;$ $c_{3}=1;$ $c_{4}=-0.411$ (**bottom**).

**Figure 10 materials-17-02738-f010:**
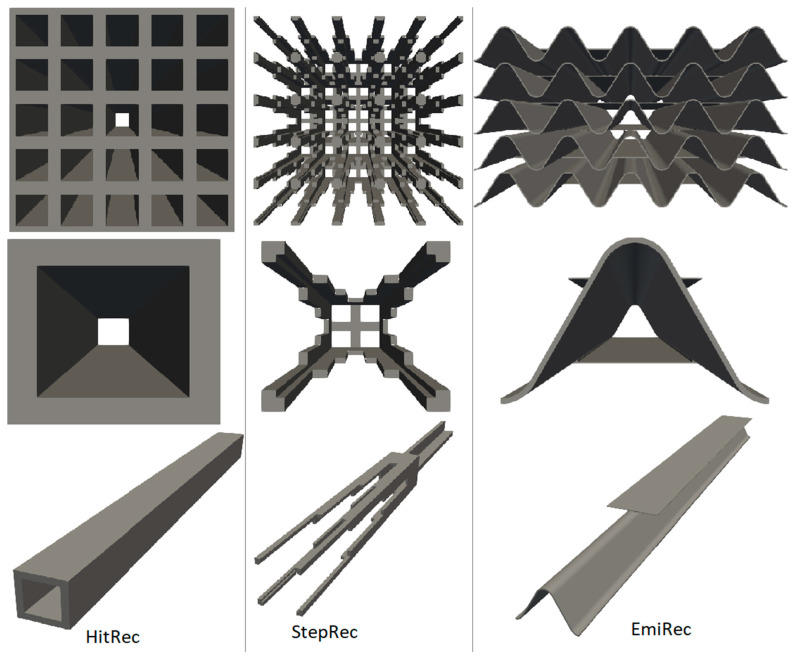
Unit channel models for the validation’s absorber geometries HiTRec, StepRec, andEmitec: a front view of 5 by 5 unit cells; a side and front view of a single unit cell.

**Figure 11 materials-17-02738-f011:**
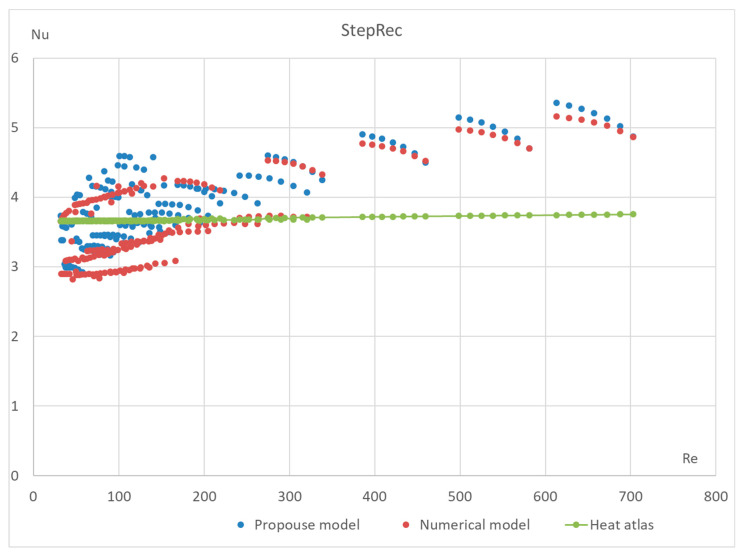
Validation of the Nu number of the dependence on Re for all sections of absorber StepRec with 35 different modes. Heat Atlas [16].

**Figure 12 materials-17-02738-f012:**
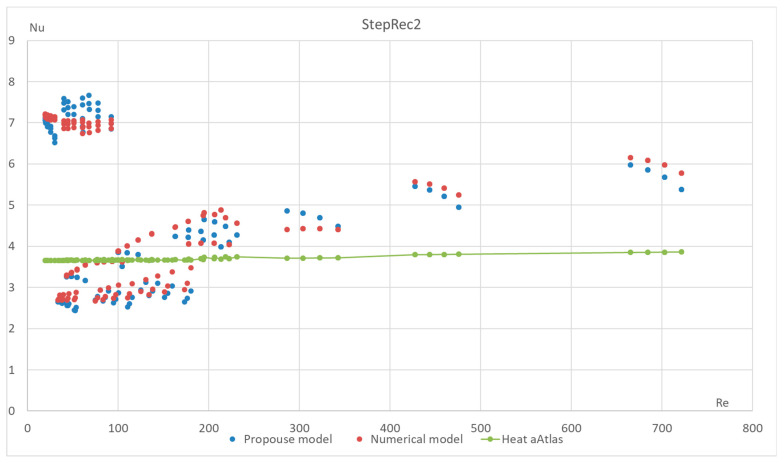
Validation of the Nu number of the dependence on Re for all sections of absorber StepRec2 with 35 different modes. Heat Atlas [16].

**Figure 13 materials-17-02738-f013:**
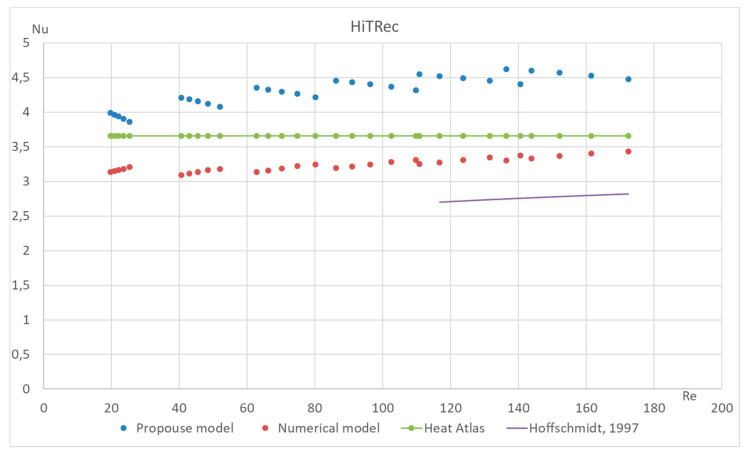
Validation of the Nu number of the dependence on Re for all sections of absorber HiTRec with 30 different modes. Heat Atlas [16], Hoffschmidt [26].

**Figure 14 materials-17-02738-f014:**
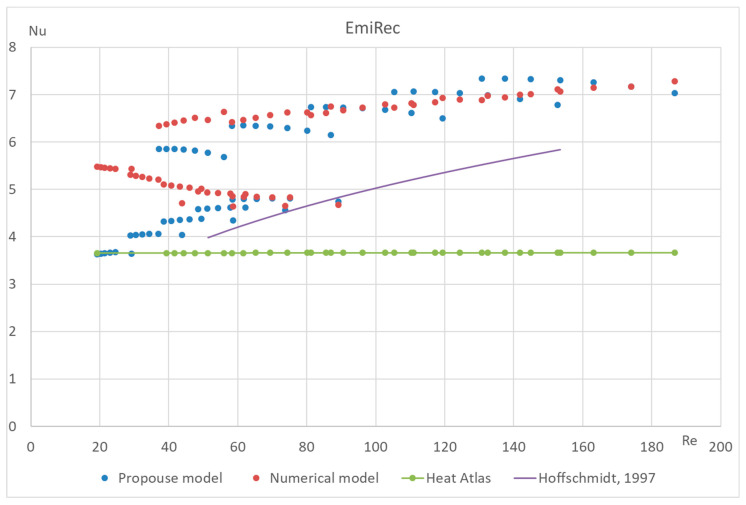
Validation of the Nu number of the dependence on Re of absorber EmiRec with 35 different modes. Heat Atlas [16], Hoffschmidt [26].

**Table 1 materials-17-02738-t001:** Calculation of the constants of the proposed equation.

Geometric Properties				Calculation					
Solar Absorber	φ	Av	de	Constant c from Statistical Data	Const. Power Coefficient n from Statistical Data	Tan((1 − φ)/φ)	Constant c1 Calculation	c1/φ	$\xi_{Shape}$
StepRec2 1 section	0.945	366.7	0.0103	0.0107	1.3919	0.058	0.010	0.035	3.187
StepRec2 2 section	0.865	733.3	0.0047	0.0132	1.324	0.157	0.011	0.039	3.112
StepRec2 3 section	0.784	1100	0.0029	0.0135	1.3276	0.283	0.012	0.042	2.851
StepRec2 4 section	0.703	1230	0.0023	0.016	1.2319	0.450	0.013	0.077	4.279
StepRec2 5 section	0.459	1987	0.0009	0.0403	1.2445	2.418	0.034	0.105	1.404

**Table 2 materials-17-02738-t002:** The statistic deviations of the invented formula.

Solar Absorber	Statistic Deviations		
	Average	In Maximum	In Minimum
StepRec2 1 section	−2.1	3.7	−6.9
StepRec2 2 section	6.5	17.2	−4.1
StepRec2 3 section	0.4	15.1	−18.3
StepRec2 4 section	−1.9	12.0	−16.1
StepRec2 5 section	0.5	12.8	−10.7
Average	0.7		

## Data Availability

Available on request.

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
