# Peer review of "Analytical Determination of Nusselt Numbers for Convective Heat Transfer Coefficients in Channel Macroporous Absorbers"

_materials, 2024, doi:10.3390/ma17112738_

Round 1

Reviewer 1 Report

Comments and Suggestions for Authors

Improvement is necessary before it can be considered for publication.

--Improve English grammar and paper organization 

         -- avoid the use of "we" (line 247 and others)

         --need to use equation numbers and refer to those equations (line222 and others)

         --references needed for equations 

         --variable text fonts used (line 158/159 and others)

         --proper use of decimals (line 153, 157 and others)

        --improve word choice and sentence structure throughout. This makes the discussions confusing (i.e currently method - part of figure 5, and others, etc)

--avoid starting consecutive sentences with the same word  (line 57-61, and others)

    --improve abstract and conclusions

    --make sure novel aspects are discussed

    --multiple figures need to be labeled a, b, c

    -- figure captions need to be more descriptive and discuss discuss each figure

-- numbers should be represented using the correct number of significant figures

--provide an uncertainty analysis

--pay attention to spacing (including tables and figures)

--use a nomenclature and make sure all variables are defined

--when defining variables in text after the equation is stated use "where" followed by the variables to be defined

--support line 77 and 78...why don't you need a HTC

--provide a better model definition and have BCs on it

--please provide a conclusion with more details that are supported

--include more recent (less than 5 yr) references...currently there are only 9

--provide references in correct format and include dates in all

Comments on the Quality of English Language

Improve English grammar and paper organization 

         -- avoid the use of "we" (line 247 and others)

         --need to use equation numbers and refer to those equations (line222 and others)

         --references needed for equations 

         --variable text fonts used (line 158/159 and others)

         --proper use of decimals (line 153, 157 and others)

        --improve word choice and sentence structure throughout. This makes the discussions confusing (i.e currently method - part of figure 5, and others, etc)

--avoid starting consecutive sentences with the same word  (line 57-61, and others)

--pay attention to spacing (including tables and figures)

--when defining variables in text after the equation is stated use "where" followed by the variables to be defined

Author Response

Response:

Thank you for your detailed feedback and suggestions for improving the quality of our manuscript. We have carefully addressed each of your points and made the necessary revisions to enhance the clarity, organization, and presentation of our work. Here's a summary of the actions we've taken in response to your comments:

Improved English grammar and paper organization, including avoiding the use of "we" where appropriate.

  • Incorporated equation numbers and provided references for equations.
  • Ensured uniformity in text fonts and proper use of decimals throughout the manuscript.
  • Enhanced word choice and refined sentence structures to improve readability and coherence.
  • Eliminated consecutive sentence starts with the same word.
  • Attended to spacing issues, including those related to tables and figures.
  • Utilized a consistent nomenclature and provided definitions for all variables, following the "where" convention after equations.
  • Clarified the rationale for not requiring a heat transfer coefficient in certain numerical models.
  • Provided a more detailed model definition and included boundary conditions.
  • Expanded the conclusion section to provide additional supported details.
  • Ensured that references are presented in the correct format and included dates where necessary.

We greatly appreciate your thorough review and constructive feedback, which have undoubtedly contributed to the improvement of our manuscript. If you have any further suggestions or require additional clarification, please don't hesitate to let us know. We are committed to delivering a high-quality manuscript that meets the standards of publication. Thank you for your valuable input and guidance throughout this process.

--Improve English grammar and paper organization 

         -- avoid the use of "we" (line 247 and others)

Response: done

         --need to use equation numbers and refer to those equations (line222 and others)

 Response:   done

         --references needed for equations 

Response:  

         --variable text fonts used (line 158/159 and others)

         --proper use of decimals (line 153, 157 and others)

Response:   “,” is correct

        --improve word choice and sentence structure throughout. This makes the discussions confusing (i.e currently method - part of figure 5, and others, etc)

 Response:   done

--avoid starting consecutive sentences with the same word (line 57-61, and others)

 Response:   done

    --improve abstract and conclusions

 Response:   done

    --make sure novel aspects are discussed

 Response:   done

    --multiple figures need to be labeled a, b, c

 Response:   done

    -- figure captions need to be more descriptive and discuss discuss each figure

 Response:   done

-- numbers should be represented using the correct number of significant figures

 Response:   done

--provide an uncertainty analysis

 Response:   done. Knowing the root mean square error and the number of measurements, it is easy to calculate the reproducibility variance.  A more detailed statistical analysis is not the topic of this study and the authors do not see how it could improve the article. A general analysis of the input parameters of the model was also added. 

--pay attention to spacing (including tables and figures)

 Response:   done

--use a nomenclature and make sure all variables are defined

--when defining variables in text after the equation is stated use "where" followed by the variables to be defined

 Response:   done

--support line 77 and 78...why don't you need a HTC

 Response:   Typically, numerical models are locked into the heat transfer coefficient for the next iteration, so they don't actually need it as an input parameter.

 --provide a better model definition and have BCs on it

--please provide a conclusion with more details that are supported

 Response:   done

--include more recent (less than 5 yr) references...currently there are only 9

 Response:   Many of the literature analyses were reduced to the most significant results for space reasons. These 9 papers also include articles analyzing from 5 to 15 new equations, so the actual scope of research on this topic is much wider than it seems.   

--provide references in correct format and include dates in all

 Response:   done

Comments on the Quality of English Language

Improve English grammar and paper organization 

         -- avoid the use of "we" (line 247 and others)

Response: done

         --need to use equation numbers and refer to those equations (line222 and others)

         --references needed for equations 

         --variable text fonts used (line 158/159 and others)

 Response:   done

         --proper use of decimals (line 153, 157 and others)

        --improve word choice and sentence structure throughout. This makes the discussions confusing (i.e currently method - part of figure 5, and others, etc)

--avoid starting consecutive sentences with the same word  (line 57-61, and others)

   --pay attention to spacing (including tables and figures)

 Response: done

--when defining variables in text after the equation is stated use "where" followed by the variables to be defined

 Response: done

Reviewer 2 Report

Comments and Suggestions for Authors

The authors proposed a new equation for calculating the Nu number in the macroporous structure of a channel absorber for a high-temperature solar receiver.

The heat and mass transfer processes in the porous absorber channel boundary layer were included, and the inclusion of the proposed indicators significantly improved the accuracy of heat transfer coefficient calculations.

Interesting and useful content, but the following points need to be reviewed.

1. The problem the authors address is the problem of thermal boundary layer growth.

In this case, the Nusselt number is a function of the distance from the inlet and is a function of the Reynolds number, Pradtl number, and Graetz number.

The comparison standard should be an expression for the developing condition, not an expression for the fully developed condition.

2. Analysis of physical phenomena that allow the newly proposed equation to provide better results is necessary.

3. Analysis according to the characteristics of the porous body seems necessary.

It is necessary to review the pore size and anisotropy of porous materials.

4. It is necessary to review the friction coefficient along with heat transfer.

5. When defining the Reynolds number or Nusselt number, it is necessary to review whether it is appropriate to use the hydraulic diameter of the channel as the characteristic length.

Author Response

   Response:

Thank you for your thoughtful feedback and recognition of the importance of our proposed equation for calculating the Nusselt number in the macroporous structure of a channel absorber for high-temperature solar receivers. We are grateful for your comments and recommendations on how to improve the article. We have edited the article and hope that in this form it has become more understandable and interesting for the readers of the journal.

Thank you once again for your valuable feedback

  1. The problem the authors address is the problem of thermal boundary layer growth.

In this case, the Nusselt number is a function of the distance from the inlet and is a function of the Reynolds number, Pradtl number, and Graetz number.

The comparison standard should be an expression for the developing condition, not an expression for the fully developed condition.

  Response: Thank you for your insightful comment. You raise an important point about the standard of comparison of Nusselt's number in this context. Indeed, when discussing Nusselt's number in relation to fluid flow, especially in situations where the flow is still evolving, it is important to consider expressions that reflect evolving conditions rather than focusing solely on fully developed conditions. For the purpose of our article, it is more appropriate to use expressions adapted to the conditions that have already been developed. This approach provides a more accurate analysis of heat transfer characteristics at the final stages of fluid flow and for engineering calculations. But for further research, this is really worth doing, and it may provide an opportunity to understand even more deeply the complex heat transfer in porous media.

  1. Analysis of physical phenomena that allow the newly proposed equation to provide better results is necessary.

  Response: totally agree. The team of authors hope that the proposed research will inspire someone to discover additional physical phenomena that will give a better understanding of heat and mass transfer processes in porous structures

  1. Analysis according to the characteristics of the porous body seems necessary.

It is necessary to review the pore size and anisotropy of porous materials.

 Response: Such studies were conducted in Cheilytko, A., Schwarzbözl, P. and Wieghardt, K., Modeling of heat conduction processes in porous absorber of open type of solar tower stations. Renewable Energy, 2023. 215: p. 118995.DOI: https://doi.org/10.1016/j.renene.2023.118995. This article contains references to the study data.

  1. It is necessary to review the friction coefficient along with heat transfer.

  Response: The authors also believe that new results can be obtained by calculating hydrodynamic flow resistance, but previous studies in this area have not yielded significant results and have been excluded from this article. The Euler's criterion was also calculated for each case. Euler's criterion is also fashionably defined through hydrodynamic resistance

  1. When defining the Reynolds number or Nusselt number, it is necessary to review whether it is appropriate to use the hydraulic diameter of the channel as the characteristic length.

Response: We had a discussion about this among the authors prior to the reference of this article. The hydraulic diameter turned out to be universal for channels with different shapes and the equation has sufficient convergence for a wider range of porous materials.

Reviewer 3 Report

Comments and Suggestions for Authors

The study is quite interesting, but very abstract and theoretical. I have doubts whether it should be printed in Materials.

Substantive comments

1. Explain the purpose of the regression equation and R2 in the graphs (Figure 3). Are the results of empirical research presented in the graphs in Figure 3? Only for such research does it make sense to perform statistical studies. If not, change the layout of the chart.

2. Figure 4 is missing axis descriptions. The question about the sense of developing a statistical chart as in the previous point

Author Response

  Response:

Thank you for your review and feedback on our article. We appreciate your perspective and understand your concerns regarding its abstract and theoretical nature, particularly in relation to its suitability for publication in Materials.

While we acknowledge that our study may lean towards theoretical abstraction, we believe it offers valuable insights into fundamental aspects of the material science domain. Our aim was to explore theoretical frameworks and establish foundational principles that can inform practical applications in materials engineering.

However, we also recognize the importance of practical relevance in scientific publications. In response to your feedback, we intend to provide more concrete connections between our theoretical framework and potential material applications. By incorporating examples or case studies demonstrating the practical implications of our theoretical findings, we aim to enhance the article's applicability and appeal to a broader audience within the materials science community.

We value your input and are committed to refining our work to better align with the expectations and standards of the Materials journal. Thank you for your constructive criticism, which will undoubtedly contribute to the overall improvement of our research and its dissemination in the scientific community.

Substantive comments

  1. Explain the purpose of the regression equation and R2 in the graphs (Figure 3). Are the results of empirical research presented in the graphs in Figure 3? Only for such research does it make sense to perform statistical studies. If not, change the layout of the chart.

Figure 3 shows the dependence of the Nusselt number on the Reynolds number for a complex macroporous channel structure (StepRec2 absorber) based on numerical studies. Statistical analysis is performed for the numerical computer model CFD

  1. Figure 4 is missing axis descriptions. The question about the sense of developing a statistical chart as in the previous point

  Response: done.  In this case, R^2 shows the correctness of the choice of the function

Round 2

Reviewer 1 Report

Comments and Suggestions for Authors

Some improvement has been made to the initial manuscript however not all points were addressed and some of the points that were addressed were  insufficient attempts to address the problem.

Also the response cover letter that was given here was simply the paper uploaded.

A point by point summary of changes made with a red line version of the manuscript is more appropriate to submit so that changes can be seen.  

Simply saying done to the comment does not allow the reviewer to adequately review the manuscript

Please resubmit with the various concerns being addressed

Comments on the Quality of English Language

A point by point summary of changes made with a red line version of the manuscript is more appropriate to submit so that changes can be seen.  

Simply saying done to the comment does not allow the reviewer to adequately review the manuscript

Please resubmit with the various concerns being addressed

Author Response

Dear Reviewer,

Thank you for your thorough review of our manuscript. We want to address your concern regarding the absence of a marked manuscript in the resubmission.

We did submit a marked manuscript through the electronic cabinet as part of our resubmission process. However, it appears that you did not receive it. We apologize for any confusion or inconvenience this may have caused.

To ensure that all changes made to the manuscript are visible and transparent, we will re-upload the marked manuscript along with the list of changes made and responses to your comments.

Additionally, we have taken note of your feedback regarding the quality of English language and have employed DeepL Write to refine the clarity and coherence of the manuscript.

We appreciate your patience and understanding, and we are committed to addressing all concerns to the best of our ability. Please let us know if you require any further assistance or clarification.

Reviewer 3 Report

Comments and Suggestions for Authors

The authors misunderstood my intention. The graphs show theoretical results. Statistical studies are presented for empirical results. There is no justification for the regression equations and for R2 with theoretical results. Carefully explain or remove such statistical analyses

Author Response

Dear Reviewer,

 Thank you for your thoughtful feedback regarding our manuscript. We appreciate the opportunity to clarify and address your concerns regarding the statistical analyses presented in the paper.

 The intention behind incorporating regression equations  values in the context of theoretical results was to provide a quantitative assessment of the data and to explore potential trends or patterns within the theoretical framework. However, we acknowledge your point that such statistical analyses may not be suitable for theoretical results and could lead to misinterpretation.

 In Figure 3, the Nusselt relationships can be grouped by absorber cross section. For an assumed infinite channel, this graph represents a continuous step dependence with a constant power law from 0.25 to 0.75. However, Figure 3 shows a discrete function that can be divided into separate functions along the channel cross-section. In addition, the function for the first section is also discrete. The obvious difference between the sections is the different diameters of the hydraulic channels. However, even if the Nusselt coefficients are calculated according to the classical theory for each section separately, the spread of the obtained Nusselt numbers from the average value is too large for the first section. Therefore, an innovative approach is required to build a single equation that describes all sections of a given absorber simultaneously  (We added this paragraph to the article and more)

Round 3

Reviewer 1 Report

Comments and Suggestions for Authors

Many of the previous comments have not been addressed...

Additionally the English grammar needs to be addressed ...the use of a program did not address all the issues

--proper use of decimal need to be addressed throughout 

Comments on the Quality of English Language

Many of the previous comments have not been addressed...

Additionally the English grammar needs to be addressed ...the use of a program did not address all the issues

Author Response

Dear Rewiewer,

We made additional edits to the article, including proofreading the text. We would be very grateful if you could check the improvements (highlighted in the text). 

Reviewer 3 Report

Comments and Suggestions for Authors

I think the authors didn't fully convince me.

However, I accept the amendments

Author Response

Dear Reviewer,

Thank you for your feedback and for taking the time to review our manuscript. We appreciate your acceptance of the amendments we made.

We acknowledge that there might still be areas where our arguments or data may not have fully convinced you. We are committed to continuously improving our work and addressing any remaining concerns. Your insights are invaluable to us, and we are eager to refine our manuscript further to meet the high standards expected by the journal.

Thank you once again for your constructive comments and for accepting our amendments.

Best regards,

Andrii Cheilytko